# Clinical Characteristics and Laboratory Findings in Children with Multisystem Inflammatory Syndrome (MIS-C)—A Retrospective Study of a Tertiary Care Center from Constanta, Romania

**DOI:** 10.3390/healthcare11040544

**Published:** 2023-02-12

**Authors:** Cristina Maria Mihai, Tatiana Chisnoiu, Adriana Luminita Balasa, Corina Elena Frecus, Larisia Mihai, Alexandru Cosmin Pantazi, Alina Zorina Stuparu, Any Axelerad

**Affiliations:** 1Department of Pediatrics, General Medicine Faculty, Ovidius University, 900470 Constanta, Romania; 2Department of Pediatrics, County Clinical Emergency Hospital of Constanta, 900591 Constanta, Romania; 3Department of Neurology, General Medicine Faculty, Ovidius University, 900470 Constanta, Romania; 4Department of Neurology, County Clinical Emergency Hospital of Constanta, 900591 Constanta, Romania

**Keywords:** COVID-19 infection, multisystem inflammatory syndrome, hyperinflammation

## Abstract

A new hyper-inflammatory syndrome in children was identified after SARS-CoV-2 infection as a post-infectious complication that is temporally associated with coronavirus disease (COVID-19). Fever, rash, conjunctival hyperemia, and gastrointestinal problems are all clinical manifestations of multisystem inflammatory syndrome in children. This condition, in some cases, causes multisystem involvement, affecting multiple organ systems and necessitating admission to a pediatric intensive care unit. Due to limited clinical studies, it is necessary to analyze the characteristics of the pathology in order to improve the management and long-term follow-up of high-risk patients. The objective of the study was to analyze the clinical and paraclinical characteristics of children diagnosed with MIS-C. The clinical study is a retrospective, observational, descriptive research work that includes patients diagnosed with MIS-C, temporally associated with coronavirus disease, and it contains clinical characteristics, laboratory data, and demographic information. The majority of patients had normal or slightly increased leukocyte counts, which were associated with neutrophilia, lymphocytopenia, and significantly elevated inflammatory markers, including high levels of C-reactive protein, fibrinogen, the erythrocyte sedimentation rate, serum ferritin, and IL 6 and elevated levels of the cardiac enzymes NT-proBNP and D-dimers, owing to the cardiovascular system involvement in the pro-inflammatory process. At the same time, renal system involvement led to raised creatinine and high proteinuria in association with hypoalbuminemia. This characteristic of the pro-inflammatory status as well as multisystem impairment are highly suggestive of the post-infection immunological reaction of the multisystem syndrome temporally associated with SARS-CoV-2 infection.

## 1. Introduction

“Kawasaki-like” syndrome was the first term because it presented clinical manifestations like Kawasaki disease. Since then, it has been referred to as multisystem inflammatory syndrome in children (MIS-C), or PIMS-TS (pediatric multisystem inflammatory syndrome temporally associated with SARS-CoV-2 infection), which typically manifests between four and six weeks after an SARS infection in a child. These terms have all been used to describe the multisystem inflammatory syndrome in children [1,2].

Three prominent explanations exist regarding the mechanisms of MIS-C and postulate an autoimmune reaction after infection. Individuals with MIS-C appear many weeks following acute infection with reduced viremia, decreased immunoglobulin M (IgM) levels, and elevated IgG and IgA levels [3]. As with other viral causes of autoimmunity, it is probable that SARS-CoV-2 initiates an autoimmune reaction that results in tissue destruction [4,5]. The prolonged presence of the SARS-CoV-2 antigen, which causes an immunological response that is not well controlled, is the subject of a second theory. Yonker et al. demonstrated that, weeks after acute infection, the SARS-CoV-2 antigen remains detectable in the digestive system of individuals who acquired MIS-C [6]. Lastly, analogous to patients experiencing toxic shock syndrome, a third explanation proposes that the “superantigen-like activity of the SARS-CoV-2 spike protein” induces an aberrant immune reaction with cytokine storming [4]. In terms of cardiovascular impairment, it is well established that the production of cytokines (Tumor necrosis factor-alpha, IL-1, IL-2, IL-6) is a recognized cause of left ventricular failure, impacting the extracellular components and myocardial cells’ force of contraction [7].

Laboratory results indicating the hyperinflammation process response include high values of C reactive protein, fibrinogen, the erythrocyte sedimentation rate, serum ferritin, and D-dimers, which are suggestive of the pathogenesis of MIS-C [2]. In some cases, coagulation disorders were described, with elevated values of D-dimers, prolonged PT, and PTT, which determines hyperfibrinogenemia [8]. It is suggested and effective to employ first-line therapy consisting of intravenous immunoglobulins (IVIG), corticotherapy, and aspirin; nevertheless, there are certain instances that do not respond to this treatment [9].

A common onset of clinical characteristics described in MIS-C is represented by fever, which is persistent for more than three days and is associated with other manifestations such as dermatological (rash, conjunctival hyperemia) or respiratory symptoms. In some cases, patients reported having digestive symptoms such as nausea, vomiting, diarrhea, or discomfort in the stomach area [10]. They also reported having cardiac involvement such as myocardial dysfunction, coronary aneurysms, or pericarditis, in addition to having elevated levels of troponin T and N-terminal pro-brain natriuretic peptide (NT-proBNP) [10]. Symptoms related to the cardiovascular system are the ones that are seen most often in patients with MIS-C. In patients with MIS-C, cardiac biomarkers including NT-pro-BNP and troponin levels are abnormally increased, which suggests heart failure and myocardial damage.

Patients may have both acute and subacute consequences of the disease, with moderate to severe cardiovascular symptoms, despite the fact that it is unclear what the long-term repercussions of MIS-C will be for the cardiovascular system. New research indicates that a host immune reaction, high levels of the activation of innate immune mechanisms, a massive increase in the proinflammatory cytokine, dysregulated thromboinflammation, thrombotic angiopathy, and endothelial dysfunction could be implicated in the development of COVID-19-related myocardial damage [11].

## 2. Materials and Methods

The research that was conducted in the Pediatrics Tertiary Care Center of the Constanta County Emergency Hospital was a retrospective, observational, descriptive study that lasted for a period of 18 months (May 2020 to October 2021). It included 35 patients who were diagnosed with MIS-C, following clinical characteristics, laboratory data, and demographic information. There were 13 male patients and 22 female patients in the study.

The statistical study was carried out using Excel and version 26 of the SPSS IBM program (v. 2010). We used Kolmogorov–Smirnoff tests, *t*-tests, and Fisher’s Exact tests in our descriptive statistics. For the purposes of the statistical analysis, the information gleaned from the data was presented in the form of absolute numbers and percentages for categorical variables. The threshold of significance was a *p* value less than 0.05.

The main objective of the study was to analyze the clinical characteristics and laboratory findings of MIS-C, considering the diagnosis, treatment, and long-term follow-up of patients.

Based on recent worldwide studies and guidelines, we suspected MIS-C and per-formed extensive laboratory tests for differential diagnosis, such as other viral and bacterial infections. Most patients had Kawasaki-like clinical manifestations at the time of admission, with persistent fever and rash.

Laboratory data were gathered, as well as evidence of SARS-CoV-2 infection, using IgG anti-SARS-CoV-2 antibodies (method: microparticle chemiluminescence, CMIA), treatment, and evolution.

Because of the large number of patients who were admitted to the department in such a short amount of time, it was decided to carry out the clinical monitoring of patients as well as laboratory tests in accordance with the recommendations that were already decided and the protocol that was established internally.

### 2.1. Consent and Ethics

In the Pediatrics Clinic of the Constanta County Emergency Hospital, the research was carried out between the months of May 2020 and October 2021. The present study was approved by the Ethics Committee of the Constanta Clinical Hospital in Romania (no.13/ 11 April 2022), and all of the parents or legal guardians were required to fill out a written informed permission form. The principles outlined in the Declaration of Helsinki were adhered to during the course of this research (ethical considerations).

### 2.2. Criteria for Inclusion and Exclusion

Participants who met the parameters of the CDC case definition for MIS-C and presented with multisystemic and hyperinflammatory states were considered for inclusion in the study. These participants had signs, symptomatology, and high laboratory results. One of the requirements for participation was the demonstration of a positive reaction to the SARS-CoV-2 IgG nucleocapsid and anti-spike antibodies.

### 2.3. Patient Information

There were 35 patients hospitalized at the Pediatric Department, ranging in age from 9 months to 15 years. Thirteen of the patients were female, making up 37% of the total, while twenty-two of the patients were male, making up 62% of the total. On the SARS-CoV-2 rapid-antigen test, the patients all had negative results. At the time of admission, 3 patients tested positive for the infection using the RT-PCR method, whereas 32 patients had negative results from the test. All of the patients tested positive for significant levels of the SARS-CoV-2 IgG nucleocapsid and anti-spike antibodies when they were put through serological testing.

## 3. Results

The onset of dermatological involvement characterized by polymorphous rash (11% females and 19% males, *p* = 0.03), conjunctival hyperemia (11% females and 19% males, *p* = 0.03), or erythematous mucosa (18% females and 26% males, *p* = 0.01) was present in 83.4% of patients (*p* = 0.01), with progressive remission within the first days of treatment. In the studied group, the days with fever ranged from 5 to 10, with a maximal temperature from 38 °C to 40.1 °C. Furthermore, patients showed improvement in inflammatory markers and system involvement in the first two weeks of treatment. There were no deaths. Additionally, the creatinine and albumin levels normalized.

In order to distinguish the diagnosis between MIS-C and Kawasaki disease, we observed the presence of thrombocytopenia, moderate lymphocytopenia, and neutrophilia in our patients. In addition, MIS-C is characterized by increased D-dimer levels (*p* = 0.01) and high NT-proBNP values (*p* = 0.02) in the laboratory. These findings are attributable to the involvement of the heart in the condition characterized by a hyper-inflammatory immune response.

Gastrointestinal symptoms represented by nausea, vomiting, and diarrhea were associated with elevated levels of fecal calprotectin in three patients.

At the onset, it was observed that most patients had normal or slightly increased leukocyte numbers (*p* = 0.02) (Figure 1A), associated with neutrophilia (*p* = 0.04) (Figure 1B) and lymphocytopenia (*p* = 0.02) (Figure 1C), thrombocytopenia (Figure 1D) and significantly elevated inflammatory markers, with high values of C-reactive protein (CRP) (*p* = 0.02) (Figure 1E), fibrinogen (*p* = 0.03), the erythrocyte sedimentation rate (*p* = 0.03) (Figure 2A), and serum ferritin (*p* = 0.05) (Figure 2B) and elevated values of the cardiac enzymes NT-proBNP (*p* < 0.001) and D-dimers (*p* = 0.01) due to the cardiovascular system involvement in the pro-inflammatory process. There was a significant correlation between high values of CRP and the presence of lower levels of serum albumin (*p* = 0.02), as well as higher levels of fibrinogen (*p* = 0.03), serum ferritin (*p* = 0.04), and D-dimer (*p* = 0.02).

The activation of immunity caused by SARS-CoV-2 infection leads to excessive cytokine production, which is referred to as a “cytokine storm.” Increased IL-6 levels were reported in eight patients as a result of this process.

During the hospitalization, two of the patients had involvement of the renal system, as shown by an elevated creatinine level. In addition, a proteinuria was detected during the urinalysis of 34.28% of the cases, and hyperalbuminemia was observed in 17 patients (48.57%). Glutamic oxaloacetic transaminase (GOT) and glutamic pyruvic transaminase (GPT) enzymes were typically increased in 63% of cases (Figure 2C,D), with GOT and GPT levels over 70 U/l. (*p* = 0.01), in parallel with the elevation of the inflammatory markers and the D-dimer, progressively decreasing after a median of 5 days of treatment, as well as high triglyceride values (Figure 3B).

For the evidence of the prothrombotic status, it was determined that 77.14% of cases (27 patients) presented modified values of fibrinogen (*p* = 0.03, median value 530,000 mg/dL) (Figure 3A), activated partial thromboplastin time (aPTT) (*p* = 0.02, median value 37.9 s), and prothrombin time (PT) (*p* = 0.02) during hospitalization.

Furthermore, significantly high values of N-terminal pro-brain natriuretic peptide (NT-proBNP) were also recorded (*p* < 0.001) (Figure 3C), suggesting cardiac impairment within MIS-C. It was observed that 29 patients (82.85%) had elevated troponin T values (*p* = 0.02), and 28 patients (80%) had increased NT-proBNP values (median 378000 pg/mL). The most common cardiac manifestation was myocarditis, which was present in 43.2% of cases.

After five days in the hospital, cardiac marker values achieved their highest point; further treatment resulted in a gradual improvement, culminating in a normalization of both the appearance of the electrocardiogram and heart function. During the hospitalization, the patients were treated with intravenous immunoglobulins (IVIG) at a dose of 2 g/kg, cortisone, aspirin, and antibiotics. The patients had a good clinical evolution while they were receiving therapy, which resulted in clinical improvement. Additionally, the inflammatory tests were reduced, with more rapidly normalizing values of C-reactive protein compared with the erythrocyte sedimentation rate and fibrinogen.

In addition, there was a rapid improvement in cardiac enzyme levels, as well as resolution of the fever and rash that had been clinically observed. The patients did not express any negative reactions to the therapy, and it was generally well-received by the patients. A total of 10 days was the typical duration of stay for patients in the hospital (range 7–21 days). Figure 4 presents an illustration of the processes and mechanisms associated with “cytokine storm,” as well as the findings of laboratory experiments. However, the need for long-term monitoring must be taken into consideration. It is still debatable whether a large dosage of acetylsalicylic acid should be used for the therapy, although doing so is essential owing to the significant risk of coronary artery dilatation in one-third of patients who have MIS-C, as shown in recent research.

## 4. Discussion

In addition to cardiac indicators, a complete blood count, an examination of liver enzymes and renal function, as well as an assessment of inflammatory marker levels are all required to be carried out as part of the preliminary laboratory investigations. The majority of patients with MIS-C seem to have an inflammatory state, as shown by neutrophilic leukocytosis, increased erythrocyte sedimentation rates, the decrease in natrium levels, high triglyceride levels, and high reactive-C protein, procalcitonin, d-dimer, and serum ferritin levels.

Laboratory markers of inflammation represent the primary characteristics, which appear to correlate with the severity of multisystem inflammatory syndrome. This is suggested by increased levels of C-reactive protein, fibrinogen, D-dimer, and serum ferritin, as well as decreased values of serum albumin.

High levels of NT-pro-BNP and troponin were detected in our investigation, as well as other studies taken from the published scientific literature [12,13,14,15]. These results imply that cardiac dysfunction is a frequent complication associated with MIS-C. Whittaker et al. discovered that 83% of patients had raised NT-pro-BNP levels, whereas 68% had elevated troponin levels [14]. Echocardiography in two dimensions must be performed to diagnose myocarditis, pericarditis, valvular anomalies, and coronary artery abnormalities (CAAs) [16,17,18]. Variations in heart function may sometimes be revealed by electrocardiograms. Using cardiac magnetic resonance imaging, Blondiaux et al. examined four patients with MIS-C who also had myocarditis. The researchers found that all of the patients had diffuse myocardial edema, which suggests that myocarditis in MIS-C is caused by an infection.

In MIS-C, immunological responses that have not entirely suppressed a continuing infection may make it possible for the intrinsic immune system to stay engaged continuously. This is due to the fact that MIS-C is a type of chronic infection. This pervasive innate immune inflammatory response is most likely caused by SARS-CoV-2’s capacity to inhibit type 1 and type 3 interferon response signaling to the adaptive immune system without impairing cytokine production.

According to the findings of our research, 77.14 percent of the patients had altered values of fibrinogen levels, activated partial thromboplastin time, and prothrombin time, all of which were strongly indicative of a prothrombotic condition (p = 0.02). Furthermore, Capone et al. [19] observed 33 patients with MIS-C who presented with high markers of inflammation, particularly fibrinogen and D-dimer concentrations, but no signs of clinical thrombosis. Although D-dimers have been shown to have elevated fibrinogen and D-dimer concentrations in children with infections and autoimmune disorders such as SARS-CoV-2 and MIS-C, it is possible that D-dimers may not have the specificity necessary for predicting deep venous thrombosis.

In addition to this, the prevalence of thromboembolic consequences in children diagnosed with MIS-C is undetermined, and there are currently no data-supported guidelines for pediatric thromboprophylaxis. In certain instances, thrombosis has been linked to coagulopathy, despite the fact that the precise mechanism responsible for this condition has not been identified. It is quite uncommon to detect significantly high levels of troponin and NT-pro-BNP in persons who have cardiac involvement, which suggests that myocardial involvement is present.

Despite the well-documented prothrombotic state and thrombotic occurrences in adults infected with SARS-CoV-2, thromboembolic comorbidities in children have not been shown to be similar [20]. In spite of the fact that the precise reason for MIS-C is unknown, researchers have hypothesized that it is caused by an abnormal immune response, which then causes the production of cytokines and organ inflammation [21]. In this regard, MIS-C has been compared to Kawasaki disease, as well as other autoimmune illnesses, all of which have been associated with a thrombogenic condition [22,23]. This comparison is based on the fact that MIS-C is thought to be an autoimmune disease.

Our findings indicate that patients with MIS-C have a hyperinflammatory state, as evidenced by increases in white blood cells, D-dimer, ferritin, and C-reactive protein, as well as fibrinogen, activated partial thromboplastin time, and prothrombin time, all of which are correlated to a hypercoagulable state [24]. It was shown that the INR levels in 46.2% of patients were between 1.5 and 2.5 and that the PT was prolonged in 62.1% of cases, even though there were no clinical symptoms of coagulation abnormalities in either group.

There have been cases of fulminant myocarditis diagnosed in children that have been reported [25,26,27]. Polymerase chain reaction testing revealed that these children were infected with SARS-CoV-2. These patients did not suffer from any other serious health conditions. It has been discovered that neonates might develop cardiac abnormalities as early as nine days of life [28].

Myocarditis is a complication that affects between fifty and seventy percent of individuals who develop MIS-C [29,30]. Son et al. found that 47% of 518 pediatric patients diagnosed with MIS-C needed vasopressor therapy, 41% showed depressed left ventricular systolic dysfunction, 12% had coronary artery aneurysm, and 3% needed extracorporeal membrane oxygenation [31]. Additionally, in a case study including 20 patients with MIS-C with cardiac dysfunction, 50% of patients exhibited reduced left ventricular activity and myocardial edema characteristic of myocarditis on cardiac magnetic resonance imaging [32]. Pouletty et al. found that severe myocarditis needed intensive care for almost fifty percent of patients diagnosed with MIS-C [12], with the likelihood of developing the condition increasing with increasing age.

It has been observed that 9–24% of individuals diagnosed with MIS-C had anomalies in the coronary arteries [9,12,13,14,15,33]. The coronary artery anomalies that impact the majority of patients manifest as either a dilatation or a small aneurysm. There have also been reports of pericarditis, pericardial effusion, and valvular regurgitation [12,13,14,15]. Electrocardiographic anomalies include prolonged PR intervals, variations to the T wave, and alterations to the ST segment.

In the majority of the studies, almost all patients completely recovered at two weeks, and there is some indication that diastolic dysfunction might remain in a small fraction of individuals six months following acute illness [19]. According to these serious results, the American Academy of Pediatrics recommends that individuals with a confirmation of MIS-C with cardiac dysfunction avoid physical activity for at least 3–6 months post-infection and that permission be acquired from a cardiologist [34].

Myocarditis may rapidly progress; thus, a diagnosis has to be made as soon as possible. In most cases, immunomodulatory therapy is a successful treatment. After a follow-up period of six weeks, an echocardiogram on the patients indicated that the great majority of them had normal activity in the left ventricle [12,13].

As part of the cytokine storm, liver involvement was observed during the MIS-C evaluation. This was accompanied by increased levels of markers for hepatocellular injury (GOT, GPT), low values of serum albumin, and higher IL-6 levels with decreased platelet values. Additionally, the presence of elevated D-dimer values was found to correlate with the liver involvement. According to the findings of Trapani and colleagues, up to 45 percent of children diagnosed with MIS-C might show signs of mild to severe liver damage [35].

Even though the cause of MIS-C may include a number of different factors, further research is necessary to determine the effects that the inflammation has on the heart and whether or not other organ systems, such as the kidneys and the liver, are also affected.

## 5. Limitations

Various limitations apply to our investigation. The study was conducted on retrospective data collected from a limited number of pediatric patients, and treatment was adjusted to each person rather than following a defined approach. Second, due to the absence of molecular analysis, our results cannot be used to make true interpretive inferences or address the fundamental pathways of MIS-C. The wide age range of patients (9 months to 15 years) and the relatively small number of children with MIS-C are considered another limitation of this research. This is a pilot study.

## 6. Conclusions

Patients with clinical manifestations of MIS-C had a negative result on the RT-PCR test for SARS-CoV-2 but had positive levels for IgG antibodies, which is an indicator of previous infection, 4–6 weeks prior to admission.

The pro-inflammatory status was evidenced by elevated levels of C-reactive protein, ESR, fibrinogen, and IL-6. This characteristic of the pro-inflammatory status, as well as multisystem impairment, is highly suggestive of the post-infection immunological reaction of multisystem syndrome, which is temporally associated with SARS-CoV-2 infection.

The clinical characteristics at the onset of the diseases included fever, mucocutaneous manifestations, rash, conjunctivitis, edema of the hands/feet, and strawberry tongue, and myocardial failure, heart rhythm disturbances, shock, gastro-intestinal manifestations, pulmonary indications, and lymphadenopathy are among the clinical characteristics of MIS-C.

The main therapeutic objective was to manage the inflammation process. Due to the cardiovascular involvement, patients needed antiplatelet agents, initially at a high dose, to reduce the incidence of cardiovascular events. Patients need to be monitored for future coronary artery dilatation, which, in some cases, may occur, despite an improvement in clinical symptoms and laboratory inflammatory tests.

Children with MIS-C require hospitalization, and some require treatment in a pediatric intensive care unit. Supportive care and treatment must be used to minimize inflammation in damaged organs and prevent them from developing permanent impairments. The long-term follow-up of patients is essential to understanding the long-term implications and prognosis for MIS-C patients.

The data that have been presented illustrate the most important laboratory findings and clinical characteristics of children who have multisystem inflammatory syndrome. These characteristics include the possible involvement of the heart, as well as injury to the liver and an effect on the renal system. This information is useful for the clinical practice of pediatricians. The post-infection immunological reaction of the syndrome that is temporally associated with SARS-CoV-2 infection in children will be appropriately evaluated after the discharge for the follow-up of long-term implications on cardiac, liver, and renal function. The multisystem impairment and pro-inflammatory status are highly suggestive for the syndrome.

## Figures and Tables

**Figure 1 healthcare-11-00544-f001:**
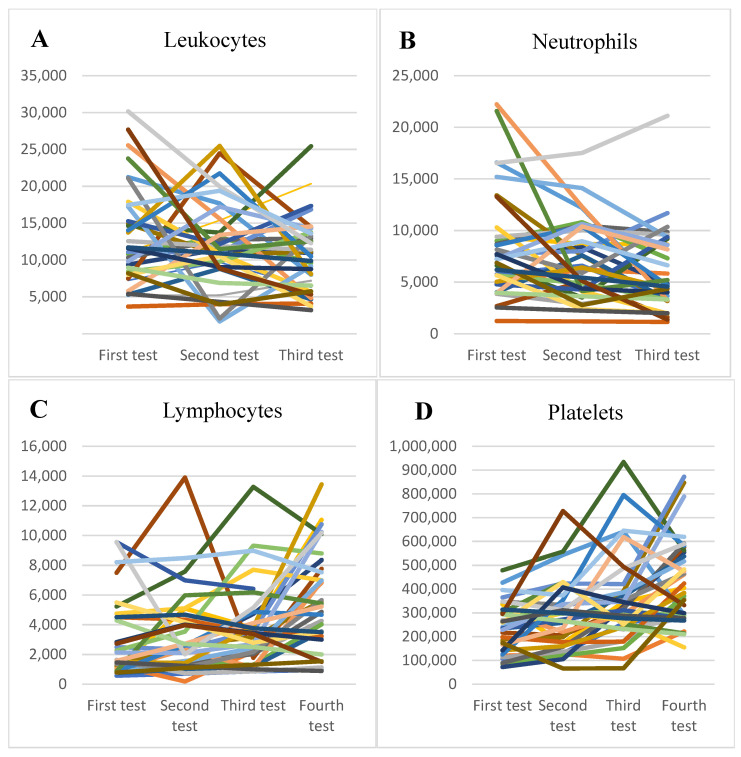
(**A**). The trendline of the values of leukocytes. (**B**). The trendline of the values of neutrophils. (**C**). The trendline of the values of lymphocytes. (**D**). The trendline of the values of platelets. (**E**). The trendline of the values of C-reactive protein.

**Figure 2 healthcare-11-00544-f002:**
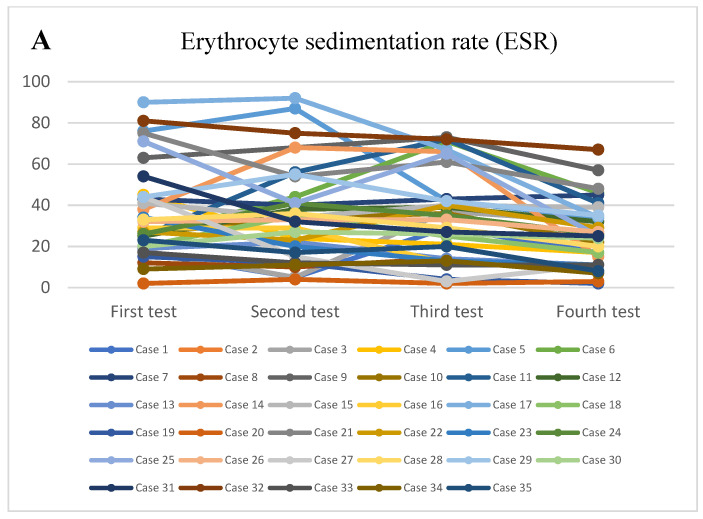
(**A**). The trendline of the values of the erythrocyte sedimentation rate. (**B**). The trendline of the values of serum ferritin. (**C**). The trendline of the values of GOT. (**D**). The trendline of the values of GPT.

**Figure 3 healthcare-11-00544-f003:**
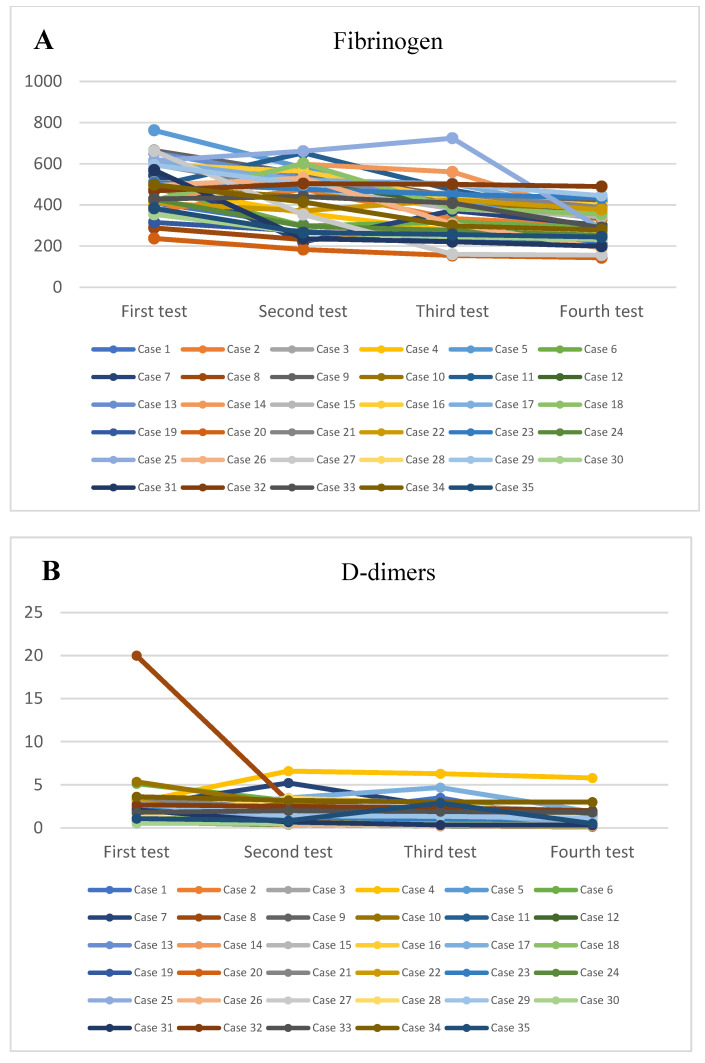
(**A**). The trendline of the values of fibrinogen. (**B**). The trendline of the values of D-dimers. (**C**). The trendline of the values of NT-proBNP.

**Figure 4 healthcare-11-00544-f004:**
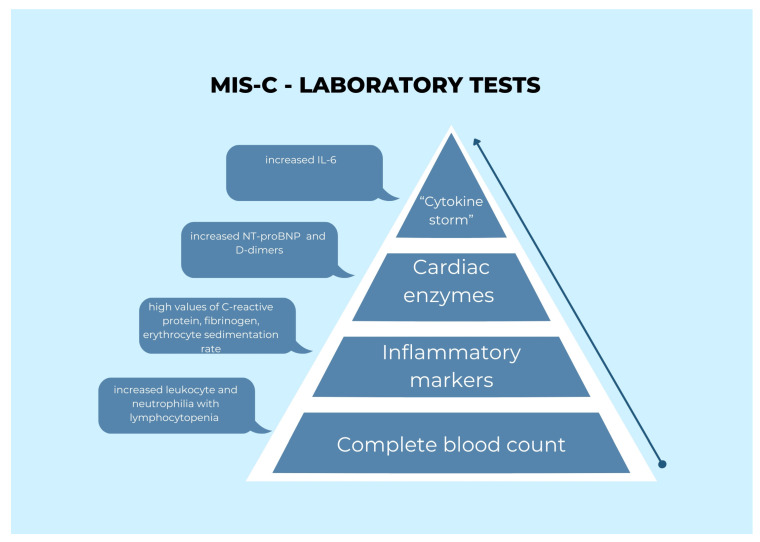
The image presented an illustration of the processes and mechanisms associated with “cytokine storm,” as well as the findings of laboratory experiments.

## Data Availability

Not applicable.

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
