# Peer review of "Clinical Characteristics and Laboratory Findings in Children with Multisystem Inflammatory Syndrome (MIS-C)—A Retrospective Study of a Tertiary Care Center from Constanta, Romania"

_healthcare, 2023, doi:10.3390/healthcare11040544_

Round 1

Reviewer 1 Report

The work deals with a very important, new and poorly known topic.

However, I have numerous methodological remarks:

1. The aim of the work should be in  the introduction and not in the methods.

2. The methods should describe all tests that have been performed in this work.

3. A history of COVID-19 infection (presence of antibodies) should be the eligibility criterion for the study- not the result  of the study (130-131)

4. The methods do not include the study of IL-6 but the result is discussed in the conclusions (276) and in Fig.4 - which is not result of this work.

5. No results of ECG, echocardiography or coronary angiography were included. However, in the discussion, the authors concluded that the ECG results improved with the normalization of cardiac enzymes (176/177).

6. Lack of information about the results of treatment (whether there were deaths or sequelae and how long was the observation of children after hospitalization.

Author Response

Thank you!

Reviewer 2 Report

Even though this study can be interesting, the main objective of the study is not clear, the study design should have enough detail to be able to be replicated. The statistical significance of the results must be reassessed, the discussion must be improved. Some points need clarification, refinement, reanalysis, rewriting, and more information to improve this article. All these points must be resolved in the manuscript before it is considered for publication.

Major points

1.       The manuscript requires a thorough correction of the wording and language editing. The title should be improved, for example, “The sFlt-1/PlGF ratio in affected pregnant patients with gestational diabetes and Sars-Cov-2 infection”. The main aim must be direct and the same throughout the manuscript (abstract, introduction, results/discussion). The first time an abbreviation appears, the full name must be entered, for example, a multisystem inflammatory syndrome in children (MIS-C). Lines 14-17: It would be a good idea to rewrite this paragraph, as "Clinical studies on MIS-C in the paediatric population are limited, and we consider it necessary to analyse its characteristics, to improve its management and long-term follow-up of high-risk patients." Authors should not use the words that appear in the title as keywords. References should be recent and relevant, they should be well referenced, and their use should be improved throughout the manuscript.

2.       The introduction section should improve. An adequate presentation of the case under study must be made. Lines 35-37: "Kawasaki-like" syndrome was the first term because it presented clinical manifestations like Kawasaki disease, and then MIS-C. Line 39-41: It was described as the evolution of a syndrome, which usually appears 4 to 6 weeks after SARS-CoV-2 infection [1], characterized by an exaggerated immune response. Line 54: IFN-? Line 61-62: Authors should avoid repeating the same information. Line 62-64: What are the authors trying to convey? Is this information relevant to establish the main aim of this study? Aside from CV damage, it would be a good idea to dig a little deeper into how other organs are affected. The research question should be clearly outlined. A good and clear justification for conducting this study should be given. It would be better if the authors offered a hypothesis before the main objective of this study. What was the main objective of this study? It should be clear what the purpose of this study was.

3.       The materials and methods sections are really sparse and require profound improvements. The authors must declare that this study was carried out in accordance with the Declaration of Helsinki (ethical considerations). The authors must describe all the variables that were evaluated. All variables studied should be described, defined, and measured appropriately. The description of the statistical analysis needs to be improved. What demographic and clinical data were compared? What Statistical Program was used? How will the results be described? This section should provide enough detail about the study design for it to be replicable. What statistical analysis and the statistical program were used to evaluate the results found? How were the results described? By percentages? Was a p-value used?

4.       The results section: In the text, the authors should describe the most significant results. It would be better if the authors subdivided this section into two or more sections: at baseline, during the course, and at discharge. It would be a good idea to use a decimal and round the numbers, for example, 37%...62.8%. How many patients had Kawasaki-type clinical manifestations? Line 123-127: This information is part of the M&M section. Lines 128-129: How did the authors assess this association? Was this association significant? Line 130-131: When was this test evaluated? The presentation of the data results is confusing. It would be better if the authors provide the results in a table. The way the authors conducted the study should be written in the M&M section. Why this result happened should be written in the discussion section. In M&M section: To demonstrate the prothrombotic state, fibrinogen levels, activated partial thromboplastin time (aPTT), and prothrombin time (PT) should be measured. All comments and suggestions should be written in the discussion section. What are the authors trying to show with Figure 4? It should be clear what were the most significant results of this study.

5.       The discussion should improve in-depth and be more argumentative. This section should start with the main objective of this study and the most significant results. Why is this study important? In this part of the manuscript, the results collected by the authors should be discussed from multiple angles and placed in context without overinterpreting them. All discussed results of the studied variables should have been described in the M&M section. A paragraph of limitations and suggestions for this study should be written before the conclusion.

6.       The conclusion must improve and be like the abstract. The introduction, the study design, and the discussion of the results should lead the reader to the same conclusion as the authors.

This manuscript is difficult to read, it appears that the results are described as a case report rather than presenting data from a retrospective study, and the discussion section does not discourse the results of this observational study. I encourage the authors to rewrite the manuscript, thinking about the principal goal of this study, and its design and answering with the results and arguments of the discussion the most proper conclusion to this research work.

Author Response

Thank you!

Reviewer 3 Report

In the reviewed article, the authors present the results of research on clinical and paraclinical symptoms in children diagnosed with MIS-C (multisystem inflammatory syndrome in children). Research on this disorder is of significant importance to pediatric practice in connection with the ongoing COVID-19 pandemic, as MIS-C symptoms are suspected to be related to Sars-Cov 2 infection. The study conducted by the authors is retrospective and descriptive in its nature. It involved a group of 35 children and adolescents in a wide age range (9 months to 15 years) diagnosed with MIS-C symptoms. This wide range of age should be considered as a limitation due to the significant differences between the infant and the adolescent. Nevertheless, the research results obtained by the authors deserve attention because they can be useful for clinical practice.

Overall, the presentation and interpretation of the results seems to be done correctly. However, from a formal point of view, the article requires some corrections and additions. First of all, part of the "Results" section should be moved to the "Materials and method" section. For example, the sentences on p. 3 (lines 104-117) refer to the method used, not to the results. In addition, in the "Limitations" section, it should be mentioned that the age range of patients (9 months to 15 years) is too wide, which is a significant limitation. Perhaps, therefore, the authors should indicate in the text that their study is of a pilot nature. In addition, on page 3 (line 99) they should replace the letters "x" and "y" with the number of male and female patients, respectively. In this context, I support the publication of the reviewed article, provided that the above-mentioned changes and additions are made in advance.

Author Response

Thank you!

Round 2

Reviewer 1 Report

The study presentation is improved now and I recommend it for publication.

Author Response

Dear Reviewer, we have made modifications of the article. Thank you for your help! 

Reviewer 2 Report

Unfortunately, the manuscript has not improved and the same gaps remain in the description of the design details and in the rest of the manuscript. The authors have not adequately responded to the suggestions presented above.

I would like to encourage the authors to respond to the suggestions given to improve the manuscript in order to be able to publish it.

Author Response

(The authors gave the same response as above.)
